# Extracellular Vesicles Derived from Human Umbilical Cord Mesenchymal Stem Cells Attenuate Mast Cell Activation

**DOI:** 10.3390/antiox11112279

**Published:** 2022-11-17

**Authors:** Tzou-Yien Lin, Tsong-Min Chang, Huey-Chun Huang

**Affiliations:** 1Department of Pediatrics, Chang Gung Memorial Hospital, Chang Gung University College of Medicine, Taoyuan 33302, Taiwan; 2Department of Applied Cosmetology, HungKuang University, Taichung 433304, Taiwan; 3Department of Medical Laboratory Science and Biotechnology, College of Medicine, China Medical University, Taichung 404333, Taiwan

**Keywords:** extracellular vesicles, umbilical cord-derived mesenchymal stem cells, mast cells, NF-κB, STAT5

## Abstract

The therapeutic potential of extracellular vesicles isolated from stem cells have been reported in several clinical diseases. Preclinical studies have demonstrated the beneficial effects of extracellular vesicles in the treatment of heart, kidney, liver, brain, and skin injuries. To address the putative therapeutic effects and mechanisms of extracellular vesicles derived from human umbilical cord mesenchymal stem cells on allergic activation in mast cells, we isolated extracellular vesicles from human umbilical cord-derived mesenchymal stem cells (UCMSCs) by tangential-flow filtration methods. The characteristics and identification of UCMSC-derived extracellular vesicles were examined via nanoparticle tracking analysis, transmission electron microscopy and protein marker analysis. Cytokines and tryptase in the cultured supernatant of KU812 cells were analyzed using an ELISA kit. Proteins in the MAPK and STAT5 signaling pathways were detected by Western blotting. This study showed that different doses of UCMSC-derived extracellular vesicles abolish IgE-stimulated KU812 cell activation and reduce the level of NF-κB, which subsequently leads to cell degranulation and the release of IL-1β, TNF-α and IL-6. Additionally, UCMSC-derived extracellular vesicles treatment blunted the IgE-induced signaling proteins p-P38, p-JNK and p-STAT5. Our results revealed a mechanism for anti-inflammation in which extracellular vesicles can affect the activation of mast cells and thus function in allergy regulation.

## 1. Introduction

Extracellular vesicles (EVs) are part of the cell “secretome”, which comprises soluble factors that participate in horizontal intercellular communication while being devoid of replicative capabilities [1,2]. The EVs originating from multivesicular endosome are enriched in several endosome-associated proteins including annexins, flotillin, GTPases, Rab as well as SNAREs [3]. Hence, some of these proteins (e.g., Tsg101 and Alix) are commonly used as markers of EVs [4]. Studies have reported that EVs package and transfer functional cargoes such as miRNAs, mRNAs, peptides, proteins, cytokines and lipids as intercellular transmitters to communicate reciprocally among cells [5,6]. In recent years, the effects of EVs on skin treatment have been extensively studied. The characteristic benefits of EVs are their high stability, nonimmune rejection, and direct stimulation of target cells [7,8]. They have also gradually become multipotent and multifunctional frontiers in dermatology and cutaneous medical aesthetics, including wound healing, skin flap reconstruction, scar removal, and skin rejuvenation [9,10,11].

Mesenchymal stem cells (MSCs) are stromal cells with multipotential. The source of MSCs includes adipose tissue, bone marrow and umbilical cord. MSCs exert minimal immunogenicity, exhibit immunomodulatory function. Additionally, MSCs possess culture expandability and multidifferentiation potential among their physiological characteristics. It has been reported that MSC-based therapies exert positive effects in the treatment of several inflammatory diseases and certain brain disorders [12,13]. The multiple beneficial properties of human umbilical cord-derived mesenchymal stem cells (UCMSCs) includes differentiation into a variety of cells of three germ layers [14], synthesis and secretion of a set of cytokines and trophic factors, supporting the expansion and function of other cells, and migration toward or home to pathological areas [15,16]. UCMSC-derived extracellular vesicles (UCMSC-EVs) have shown outstanding therapeutic effects on many organs, such as the heart, kidney, liver, some immune and neurological disorders, as well as cutaneous wound healing and anti-inflammatory properties [17]. Additionally, UCMSCs grow faster (~4-day doubling time) than mesenchymal stem cells from bone marrow or adipose tissue (~7-day doubling time). UCMSCs yielded four times as many extracellular vesicles per cell as did mesenchymal stem cells from bone marrow or adipose tissue [18]. Because of their bioactive advantages, UCMSC-EVs are likely to become a promising new approach for tissue repair and regeneration [19].

Human skin-resident mast cells respond to microbial-derived products or other exogenous insults. Immune mast cells are stimulated in an allergen- and immunoglobulin E (IgE)-dependent manner and subsequently provoke allergic reactions by secreting histamine, proteases, and chemotactic factors [20]. The primary mechanism of mast cell activation involves prompted signaling of the Mitogen-activated protein kinase (MAPK) cascade, consequently leading to the activation of NF-κB, which acts as a mediator of pro-inflammatory gene TNF-α and IL-6 induction in mast cells [21]. In addition, the signal transducers and activators of transcription (STAT) pathway is a canonical cytokine-stimulated signaling pathway in inflammatory diseases. STAT5 transcription factors are critical for mast cell development and IgE-induced degranulation [22,23]. To determine the effects of UCMSC-EVs on mast cells, we first applied the dinitrophenol (DNP) IgE to sensitize KU812 cells. Second, we used DNP-human serum albumin antigen to activate the sensitized mast cells in vitro. At last, we explore the molecular mechanisms underlying the effects of UCMSC-EVs on IgE-mediated activation of KU812 cells.

## 2. Materials and Methods

### 2.1. Cell Culture, Conditioned Medium Preparation and UCMSC-EVs Isolation

Human UCMSC were provided by Chang Gung Memorial Hospital, Taoyuan, Taiwan. KU812 cells (ATCC-CRL-2099, BCRC60502) were obtained from the Bioresource Collection and Research Center (BCRC), Taiwan. Human UCMSC were cultured to 90% confluence in α-MEM containing 5% human platelet lysate and maintained in 10-layered Cell Factory Systems (Cell Factory™ Systems, CF10) at 37 °C, 5% CO_2_ and 1% O_2_. The culture medium was harvested and filtered through a 0.22 µm polyethersulfone membrane filter (Thermo Fisher Scientific, Waltham, MA, USA) to remove large particles and cell debris. Finally, EVs were isolated using a tangential flow filtration (TFF) (Sartorius Stedim Biotech, Göttingen, Germany) system with a 300 kDa molecular weight cutoff filter. The cells were further diafiltrated with 10× volumes of PBS to remove **non-EV** proteins, nutrients, and cellular waste products such as lactate and ammonia. All operations were performed at 4 °C. The final exosome pellets were stored at −80 °C. The protein content of EVs was measured by bicinchoninic acid protein assay (BCA; Thermo Fisher Scientific, Waltham, MA, USA).

The KU812 cells were cultured in RPMI-1640 medium containing 10% of heat inactivated fetal calf serum (FCS) and 1% of antibiotic antimycotic solution (penicillin/streptomycin/Fungizone), the cells were grown at 37 °C in a humidified incubator containing 5% CO_2_. The KU812 cells were first treated with DNP-IgE (100 ng/mL) for 16 h, then stimulated with DNP-HAS antigens (100 ng/mL) for 4 h, and further incubated with UCMSC-EVs at 37 °C for 24 h. The trypan blue exclusion cell number were counted via a hemocytometer.

### 2.2. Measurement of EVs Size and Concentration Distribution with Nanoparticle Tracking Analysis (NTA)

The size distribution and concentration of isolated UCMSC-EVs suspensions were analyzed using nanoparticle tracking analysis (NTA) (NanoSight NS300). The samples were diluted in PBS and mixed well, and the diluted samples were then injected into the laser chamber, yielding particle concentrations in the region of 10^6^–10^9^ particles/mL in accordance with the manufacturer’s recommendations. All samples were analyzed in triplicate.

### 2.3. Transmission Electron Microscopy (TEM)

UCMSC-EVs samples were fixed with 2.5% glutaraldehyde at 4 °C overnight. After washing, vesicles were loaded onto formvarcarbon-coated grids, negatively stained with aqueous phosphotungstic acid for 60 s and imaged with a transmission electron microscope at 80 kV (H7500 TEM).

### 2.4. Reactive Oxygen Species (ROS) Measurement

KU812 cells were seeded in 96-well plates at a concentration of 1 × 10^5^ cells/mL. Cells first sensitized with DNP-IgE (100 ng/mL) for 16 h then added with DNP-HSA (100 ng/mL) for 4 h. Subsequently, cells were treated with UCMSC-EVs for 8 h and loaded with 10 μM 2, 7-dichlorofluorescein diacetate (DCFH-DA; Sigma-Aldrich, Darmstadt, Germany) for 30 min. DCF fluorescence was assessed with a spectrofluorophotometer (Thermo Fisher Scientific Inc., Waltham, MA, USA) at the excitation (485 nm) and emission (530 nm) wavelength.

### 2.5. Cytokine Measurement

The concentrations of interleukin IL-1β, IL-6 and tumor necrosis factor (TNF)-α in the supernatants obtained from cell culture were analyzed using an enzyme-linked immunosorbent assay (ELISA) kit according to the manufacturer’s instructions.

### 2.6. Western Blot Analysis

Cells were lysed in a proteinase inhibitor containing lysis buffer at 4 °C for 20 min. The protein lysis solution contains 0.1% sodium dodecyl sulfate (SDS), 0.5% sodium deoxycholate, 1% nonidet P-40, 1 μg/mL pepstatin A, 5 μg/mL aprotinin, 100 μg/mL phenylmethyl sulfonyl fluoride and 1 mM ethylenediaminetetraacetic acid (EDTA). Proteins (10 μg) were resolved by SDS-polyacrylamide gel electrophoresis and electrophoretically transferred to a polyvinylidene fluoride (PVDF) membrane. The membrane was blocked in 5% fat-free milk in PBST buffer followed by overnight incubation with the following primary antibodies diluted in PBST: p-38 (1:2500, Cell Signaling Technology, Danvers, MA, USA), p-JNK (1:1500, Cell Signaling Technology), p65 Ab (1:1000, Abcam, Waltham, MA, USA), pSTAT5a Ab (1:1000, Abcam, Waltham, MA, USA), CD9 (1:500, System Biosciences, Palo Alto, CA, USA), CD63 (1:500, System Biosciences, Palo Alto, CA, USA), CD81 (1:500, System Biosciences, Palo Alto, CA, USA), calnexin (1:500, Abcam, Waltham, MA, USA), GAPDH Ab (1:10,000, Santa Cruz, CA, USA), and β-actin (1:10,000, Santa Cruz). A subsequent incubation with secondary antibody (1:20,000) was conducted at room temperature for 2 h. Again, the membrane was washed in PBST buffer to remove the excessive secondary antibodies. The blot was visualized using an enhanced chemiluminescence (ECL) reagent (GE Healthcare, South Jakarta, Indonesia). The relative amounts of the expressed proteins compared to the GAPDG content were analyzed using Image J software 1.53t (Bethesda, MD, USA).

### 2.7. Degranulation Assay

The KU812 cells (1 × 10^6^ cells/mL) were treated with DNP-IgE, DNP-HAS and **UCMSC-EVs** as decribed above. To measure the tryptase (a biomarker of mast cell degranulation) released from cells, a mast cell degranulation assay kit (IMM001, Millipore, MA, USA) was applied to determine tryptase levels in cell culture supernatants and lysates. To measure tryptase release (a biomarker of degranulation) from cells, supernatants and cell lysates were incubated in a substrate solution (1.3 mg/mL tosyl-gly-pro-lys-pNA in 0.1 M sodium carbonate) at 37 °C for 1 h, and the reaction was terminated by adding a stop solution (50 mM sodium carbonate) for 15 min at room temperature. Absorption was measured at 405 nm, and the tryptase release were expressed as: tryptase of [supernatant/supernatant + cell lysate] × 100%.

### 2.8. Statistical Analysis

The statistical analysis of the experimental data was performed by one-way analysis of variance (ANOVA), which was used for the comparison of measured data using SPSS 21.0 statistical software (SPSS Inc. Chicago, IL, USA). The *p* values of the data were compared with the control group and calculated by Student’s *t* test. Differences were considered as statistically significant at * *p* < 0.05, ** *p* < 0.01, *** *p* < 0.005.

## 3. Results

### 3.1. Physiochemical Properties of UCMSC-EVs

The shapes and size distributions of UCMSC-EVs were observed through TEM and NTA. The particle size analysis confirmed that the main peak of particle size was among the typical size arrangements of extracellular vesicles (30–150 nm), whereas the diameter of extracellular vesicles was approximately 103.3 ± 1.0 nm according to NTA results (Figure 1A). TEM is particularly useful to characterize the content of extracellular vesicles with the advantage of being label-free. TEM revealed that UCMSC-EVs consisted of spherical double membrane-bound vesicles (Figure 1B) and presented the exosome-positive markers CD9, CD63, CD81, Alix and Tsg101 and no contamination of the nonexosomal markers calnexin, annexin V and β-actin by immunochemical analysis (Figure 1C). The amount of UCMSC-EVs isolated by this procedure was equivalent to approximately 2 × 10^10^ particles/mg total proteins.

### 3.2. Viability of KU812 Cells under UCMSC-EVs Treatment

To examine the effects of the UCMSC-EVs on mast cells, nonadherent KU812 cells were stained with a trypan blue exclusion test to quantify the viable cells. Cultures were seeded with a total number of UCMSC-EVs ranging from 1 × 10^7^ (equivalent to 1 μg/mL total protein) to 1 × 10^9^/mL (equivalent to 100 μg/mL total protein). KU812 cells remained viable during a period of time from 24 to 72 h. There was no significant difference in the percentage of cell viability at each dose, indicating that UCMSC-EVs were not cytotoxic to KU812 cells under various challenges (Figure 2).

### 3.3. UCMSC-EVs Attenuated IE-Induced ROS Generation and Degranulation in KU812 Cells

We probed the levels of ROS in IgE-treated KU812 cells with the superoxide indicator DCFHDA. As shown in Figure 3A, marked augmentation of green fluorescence manifested the increase in intracellular superoxide levels compared to that in the control group. The addition of UCMSC-EVs resulted in a significant decrease in ROS fluorescence (approximately 50%). The application of 10^7^–10^9^ particles/mL extracellular vesicles did not show a dose–response effect on eliminating ROS.

The percentage of mast cell degranulation was significantly increased (by 350%) in response to treatment with IgE. After incubation with 10^7^–10^9^ particles/mL UCMSC-EVs for an additional 24 h, a significant decrease in degranulation was observed, further revealing the anti-inflammatory properties of the extracellular vesicles (Figure 3B).

### 3.4. UCMSC-EVs Exposure Decreases the Production of Multiple Proinflammatory Cytokines in KU812 Cells

We investigated whether UCMSC-EVs treatments could modulate the levels of the proinflammatory cytokines IL-1β, TNF-α and IL-6. The expression of IL-1β, TNF-α, and IL-6 in KU812 cells was significantly decreased in the UCMSC-EVs groups compared to the IgE-induced upregulation of these proinflammatory cytokines (Table 1).

### 3.5. UCMSC-EVs Inhibited Inflammatory and Allergic Reactions by Suppressing the NF-kB and MAPK Signaling Pathways

It has been reported that the activation of MAPK is crucial for thetranscriptional regulation of allergic responses [24]. Further, we investigated whether the inhibition of allergic responses by UCMSC-EVs is mediated through the NH2-terminal kinase (JNK) and p38 MAPK pathways in KU812 cells. As shown in Figure 4, IgE induced the activation of p38 MAPK and JNK. UCMSC-EVs suppressed IgE-stimulated p38 MAPK and JNK in KU812 cells. NF-κB is a kind of dimeric transcription factors that participates in modulation of innate and adaptive immunity. The results revealed that the protein levels of NF-κB were increased after IgE-driven KU812 cell activation, while the levels of NF-κB were reduced at a relatively low dose of 10^7^/mL UCMSC-EVs (Figure 4 and Appendix A).

The hyperactivated STAT5 is reported to promote hypersensitivity through the aberrant expression of proinflammatory genes. It is also reported that the STAT signaling pathways can cross-talk with the NF-κB and MAPK pathways. Hence, we next investigated the effects of UCMSC-EVs on the phosphorylation of STAT5 kinases, which are the most important crucial kinases responsible for the activation of mast cells. The results revealed that the UCMSC-EVs treatment apparently suppressed the phosphorylation levels of STAT5 in the activated KU812 cells.

## 4. Discussion

The ultrafiltration method may contribute most to EV**s** purification due to its advantages of easy handling, high-throughput isolation of biological samples, and capability of isolating defined sizes of extracellular vesicles with high purity. Herein, we use ultrafine nanomembranes with different molecular weight cutoffs to differentiate **UCMSC-EVs** and co-vesicles by tangential flow filtration. Ultrafiltration-based isolation of EV**s** dramatically shortens the processing time. Particularly, this method does not require any special equipment and could obtain higher recovery of exosome yields [25]. We believe optimal isolation techniques will help investigators address many of the challenges faced in **EV**-related basic and applied biomedical applications.

Mast cells are long-lived tissue-resident immune cells that are situated as first responders against external pathogens and environmental stimuli. Commonly, increased mast cell numbers could be a driving mechanism in mast cell-associated disorders [26]. The administration of **UCMSC-EVs** did not show any cytotoxicity in KU812 cells, suggesting that these concentrations are within safety limits. Administration of 10^9^ particles/mL **UCMSC-EVs** showed a gentle effect (<1.2-fold of control) on the proliferation of KU812 cells, possibly due to extracellular vesicles derived from mesenchymal stem cells promoting cell proliferation [27,28].

Previous reports have demonstrated the important roles of mast cells in the mediation of allergic reaction through the degranulation process, which is predominantly resulted from the antigen-IgE antibody reaction. Tryptases are the most abundant granule proteins (up to 25% of total cellular protein content) in human mast cells [29]. The results of the present study showed that **UCMSC-EVs** have the ability to suppress the release of tryptase and can be considered a pharmacological target for addressing the mast cell stabilizing effect and reducing IgE-mediated mast cell degranulation.

Another effect of **UCMSC-EVs** potentially involved in the downregulation of mast cell activity might be abrogating both MAPKs and STATs in the mast cell model. Our study identified that STAT5 is involved in the mechanism integrating anti-inflammatory activity of **UCMSC-EVs**. In mast cells, IgE is reported to be crosslinked with cognate receptors and activate the subsequent JAK-STAT signaling pathway [30]. The activated STATs bind to specific DNA sequences and initiate the transcription of specific target genes. STAT5 has been recognized as a key factor in those signaling pathways that control mast cell function and the expansion of the mast cell population [31]. The results of this study revealed that **UCMSC-EVs** significantly attenuated IgE-induced STAT5 phosphorylation in a concentration-dependent manner, which could be one of the main mechanisms underlying the protective effect of **EVs** on abnormal allergic inflammation and immediate hypersensitivity.

Oxidative stress plays a tremendous role in many inflammatory processes. ROS-mediated oxidation of upstream kinases can influence NF-κB, which further regulates cytokine expression in innate immunity. Furthermore, ROS also modulate STAT activation in normal and cancer cells, and in turn, STAT5 modulates ROS production, ultimately leading to the feed-forward loop that augments STAT5 activation and drives ROS formation [32,33]. These results indicate that NF-κB and ROS-mediated signaling play important roles in IgE-mediated mast cell activation [34]. Our study also demonstrated that **UCMSC-EVs** alleviated the ROS level and restrained the NF-κB pathway, suggesting that **UCMSC-EVs** could help manage allergic disorders. Notably, similar studies have shown that adipose tissue-derived mesenchymal stem cell-derived exosomes could reduce pathological symptoms in an atopic dermatitis (AD) mouse model [35]. These findings indicate that EVs could be a novel cell-free therapeutic strategy to treat skin inflammatory diseases.

## 5. Conclusions

In the present study, transmission electron microscopy analysis and nanoparticle tracking analysis revealed the size distribution and number of UCMSC-EVs. The characteristics of UCMSC-EVs were also validated by Western blotting. We provide evidence that UCMSC-EVs could act as a mast cell stabilizer by repressing IgE-dependent tryptase release. Moreover, UCMSC-EVs reduced proinflammatory cytokine production in DNP-IgE-sensitized KU812 cells by suppressing the STAT5 and NF-κB signaling pathways in a dose-dependent manner.

## Figures and Tables

**Figure 1 antioxidants-11-02279-f001:**
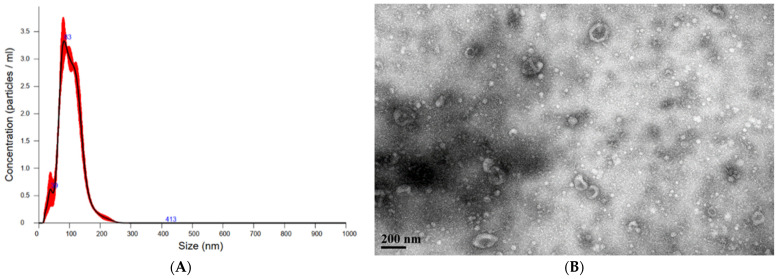
Characteristics of UCMSC-EVs. (**A**) Size distributions of UCMSC-EVs based on NTA measurements. The mean diameters were 103.3 ± 1.0 nm. (**B**) EVs were negatively stained and viewed by transmission electron microscopy at a magnification of 60,000. The scale bar represents 200 nm. (**C**) Representative immunoblot showing the expression of the EVs’ markers CD9, CD63, CD81, Tsg101 and Alix, whereas HSP70, calnexin, annexin V and β-actin acted as negative control.

**Figure 2 antioxidants-11-02279-f002:**
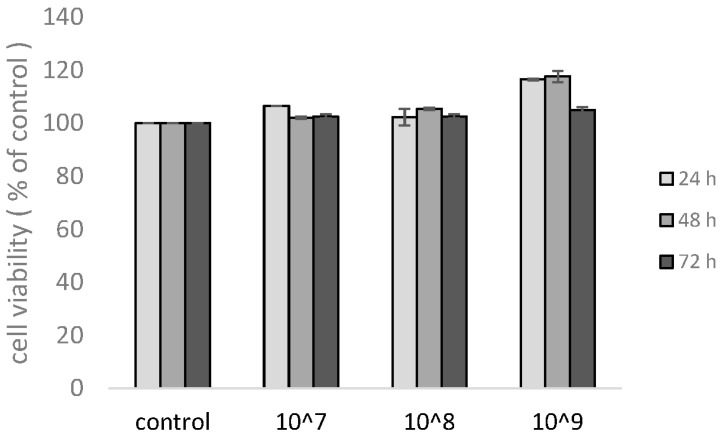
Effect of different doses of UCMSC-EVs on the viability of KU812 cells. Cells were incubated with different concentrations of extracellular vesicles for the indicated times. The number of KU812 cells was measured by trypan blue exclusion method. The results show the fold change relative to the control. Data represent the mean ± SD of three independent experiments.

**Figure 3 antioxidants-11-02279-f003:**
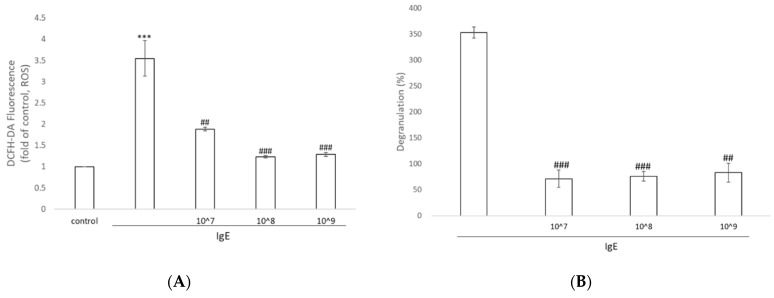
Effects of UCMSC-EVs on the generation of oxidative stress and degranulation. (**A**) Representative levels of intracellular ROS in KU812 cells stimulated with DNP-IgE or with UCMSC-EVs. (**B**) KU812 cells were either sensitized with DNP-IgE or further treated with different amount of UCMSC-EVs. The amount of tryptase released into the supernatants from 1 × 10^6^ cells was quantified using a degranulation kit after 24 h of incubation. Data represent the mean ± SD of three independent experiments. *** *p* < 0.001 compared with the untreated control and ## *p* < 0.01, ### *p* < 0.001 compared with DNP-IgE.

**Figure 4 antioxidants-11-02279-f004:**
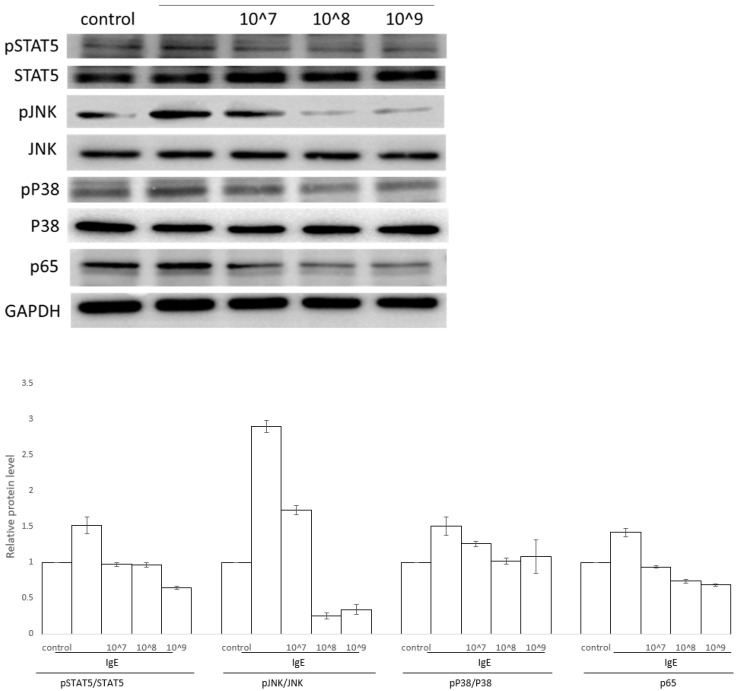
Functional mechanisms of UCMSC-EVs on DNP-IgE-activated KU812 cells. Representative Western blots of p-STAT5, p-p38, p-JNK, and p65 expression in KU812 cells treated with DNP-IgE or plus different concentrations of UCMSC-EVs. The relative fold change in phosphorylated protein to total protein levels was obtained from 3 individual datasets.

**Table 1 antioxidants-11-02279-t001:** Expression of inflammatory cytokine in the culture medium of KU812 cells.

		IgE			
pg/mL	Control		10^7^	10^8^	10^9^
IL-6	15.01 ± 1.12	41.66 ± 1.03 ^b^	26.23 ± 0.26 ^e^	20.44 ± 0.34 ^e^	24.04 ± 1.29 ^d^
TNF-α	14.08 ± 0.56	24.02 ± 0.32 ^b^	14.48 ± 0.16 ^e^	13.46 ± 0.32 ^e^	14.85 ± 0.24 ^e^
IL-β	145.10 ± 0.24	237.77 ± 11.73 ^a^	167.03 ± 4.74 ^e^	146.41 ± 8.36 ^d^	154.19 ± 22.26 ^c^

Fold of control, ^a^ *p* < 0.01, ^b^ *p* < 0.001, Fold of IgE, ^c^ *p* < 0.05, ^d^ *p* < 0.01, ^e^ *p* < 0.001.

## Data Availability

Data is contained within the article and Appendix A.

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
