# Peer review of "Extracellular Vesicles Derived from Human Umbilical Cord Mesenchymal Stem Cells Attenuate Mast Cell Activation"

_antioxidants, 2022, doi:10.3390/antiox11112279_

Round 1

Reviewer 1 Report

In this paper, authors investigated the effect of exosomes derived from human umbilical cord-derived mesenchymal stem cells on allergic activation of mast cells. After isolation and characterization of exosomes, they demonstrated that different doses of UCMSC-derived exosomes have an anti-inflammatory effect in the allergic processes because they can affect the activation of mast cells.

Previously, it has already been demonstrated the effect of MSC-derived exosomes on the activation of mast cells dose and time-dependent and their anti-inflammatory effect. In particular, Cho BS et al. demonstrated that ASC-exosomes reduced pathological symptoms such as clinical score, the levels of serum IgE, the number of eosinophils in the blood, and the infiltration of mast cells, CD86+, and CD206+ cells in skin lesions. Furthermore, ASC-exosomes also significantly reduced mRNA expression of various inflammatory cytokines such as interleukins and tumor necrosis factor-α (TNF-α) in AD skin lesions (PMID: 29996938).

In 2014 the Executive Committee of the International Society for Extracellular Vesicles (ISEV), a group of scientists with collective long-term expertise in the field of EV biology,  proposed a series of criteria, based on current best-practice, that represent the minimal characterization of EVs that should be reported by investigators. Adoption of these criteria should aid researchers in planning studies as well as reporting their results. In addition, they suggested appropriate controls that should be included in EV-related functional studies. These controls should support conclusions regarding the functions of EVs and their relationship to physiologic and pathologic mechanisms.

In the absence of any of the above-proposed controls, investigators may still conclude that an extracellular functional activity exists and affects recipient cells, but the specific EV nature of this function should not be claimed (PMID: 25536934). Moreover, in the “Introduction” paragraph, the authors cited an article about proteins that are commonly used as markers of exosomes (Tsg101 and Alix). I suggest that they could investigate these proteins to further confirm that the isolated EVs are exosomes and also to perform an Annexin V assay to confirm that there is no contamination of apoptotic bodies.

In paragraph 2.1 and in paragraph 3.3 authors discuss the results of TEM and DCFDA assay that are not mentioned in “Materials and methods”.

Paragraph 3.5 is missing references.

In Figure 4 the authors showed the Western Blot analysis. Are the numbers below quantitative data? If Yes, the authors should be shown the statistic analysis with other gels that they showed in the supplementary file.

The English writing should be improved also correcting several typing errors, including missing spaces between words and repeated sentences (for example, lines 119-123).

Author Response

Dear reviewers of Antioxidants

Thank you for reviewing our manuscript entitled “Extracellular Vesicles Derived from Human Umbilical Cord Mesenchymal Stem Cells Attenuate Mast Cell Activation.” Our revisions in response to the reviewers’ comments and concerns are addressed below in a point-by-point manner accordingly. We appreciate the time and effort that the reviewers have dedicated to providing valuable feedback on our manuscript. We are looking forward to your positive decision on our article.

Reviewer 2 Report

This is a concise, well-written article on the in vitro immunomodulatory effects of Human Umbilical Cord Mesenchymal Stem Cells on human mast cells.

The paper includes a few descriptive figures. Conclusions are supported by the afforded experimental data.

The style of the References has to be carefully revised and presented  in accordance with Journal guidelines. In particular, some references are incompletely described.

Author Response

(The authors gave the same response as above.)

Round 2

Reviewer 1 Report

Thanks to the authors for answering the questions.

You need just a small change to make to figure 4. It would be advisable to delete the "fold changes number" and create a histogram graph by adding standard deviation and statistical analysis.

Author Response

Response to Reviewer 1:

You need just a small change to make to figure 4. It would be advisable to delete the "fold changes number" and create a histogram graph by adding standard deviation and statistical analysis.

Response: Thanks to the reviewers for pointing out this issue. We have added the quantitative graph in the Fig.4.